# A Modified Basophil Activation Test for the Clinical Management of Immediate Hypersensitivity Reactions to Paclitaxel: A Proof-of-Concept Study

**DOI:** 10.3390/cancers15245818

**Published:** 2023-12-13

**Authors:** Marilena La Sorda, Marco Fossati, Rosalia Graffeo, Manuela Ferraironi, Maria Cristina De Rosa, Alexia Buzzonetti, Benedetta Righino, Nicole Zampetti, Andrea Fattorossi, Eleonora Nucera, Arianna Aruanno, Gabriella Ferrandina, Adriana Ionelia Apostol, Alessandro Buonomo, Giovanni Scambia, Maurizio Sanguinetti, Alessandra Battaglia

**Affiliations:** 1Microbiology Unit, Fondazione Policlinico Universitario A. Gemelli IRCCS, Università Cattolica del Sacro Cuore, 00168 Rome, Italy; marilena.lasorda@policlinicogemelli.it (M.L.S.); rosalia.graffeo@policlinicogemelli.it (R.G.); maurizio.sanguinetti@policlinicogemelli.it (M.S.); 2Cytometry Unit, Fondazione Policlinico Universitario A. Gemelli IRCCS, 00168 Rome, Italy; marco.fossati@policlinicogemelli.it (M.F.); alexia.buzzonetti@policlinicogemelli.it (A.B.); n.zampetti@irbm.com (N.Z.); andrea.fattorossi@policlinicogemelli.it (A.F.); 3Institute of Chemical Sciences and Technologies ‘‘Giulio Natta’’ (SCITEC)-CNR, 00168 Rome, Italy; mariacristina.derosa@cnr.it (M.C.D.R.); benedetta.righino@scitec.cnr.it (B.R.); 4Allergy Unit, Fondazione Policlinico Universitario A. Gemelli IRCCS, 00168 Rome, Italy; eleonora.nucera@policlinicogemelli.it (E.N.); arianna.aruanno@policlinicogemelli.it (A.A.); alessandro.buonomo@policlinicogemelli.it (A.B.); 5Gynecology Oncology Unit, Fondazione Policlinico Universitario A. Gemelli IRCCS, Università Cattolica del Sacro Cuore, 00168 Rome, Italy; gabriella.ferrandina@policlinicogemelli.it (G.F.); scambia@policlinicogemelli.it (G.S.); 6Department of Woman, Child and Public Health, Fondazione Policlinico Universitario A. Gemelli IRCCS, 00168 Rome, Italy; adrianaionelia.apostol01@icatt.it; 7Department of Life Science and Public Health, Università Cattolica del Sacro Cuore, 00168 Rome, Italy

**Keywords:** basophils, BAT, flow cytometry, hypersensitivity, MRGPRX2, platinum, taxane

## Abstract

**Simple Summary:**

Anaphylaxis to paclitaxel requires desensitization protocols for safe drug re-administration. Hypersensitivity severity and skin tests (STs) guide desensitization protocol choices. The only study reporting on the basophil activation test (BAT) in diagnosing hypersensitivity to paclitaxel showed unsatisfactory diagnostic performance. Here, we present a modified BAT that shows unprecedented sensitivity (90.91%) and specificity (90.91%) in diagnosing hypersensitivity to paclitaxel and demonstrate that most patients with hypersensitivity to paclitaxel who score ST-negative score BAT-positive. We propose that the present BAT should be used to personalize patient management. Moreover, we predict the interactions of paclitaxel with the degranulation-competent MRGPRX2 receptor using an in silico molecular docking study.

**Abstract:**

Immediate hypersensitivity reactions (iHSRs) to taxanes are observed in 6% and 4% of gynecologic and breast cancer patients, respectively. Drug desensitization is the only option, as no comparable alternative therapy is available. Surfactants in the taxane formulation have been implicated in the immunopathogenesis of iHSRs, although sporadic skin test (ST) positivity and iHSRs to nab-paclitaxel have suggested the involvement of the taxane moiety and/or IgE-mediated pathomechanisms. In vitro diagnostic tests might offer insights into mechanisms underlying iHSRs to taxanes. The aim of the present study was to address this unmet need by developing a novel basophil activation test (BAT). The study included patients (*n* = 31) undergoing paclitaxel/carboplatin therapy. Seventeen patients presented with iHSRs to paclitaxel (iHSR-Tax^pos^), and eleven were tolerant (iHSR-Tax^neg^). Fourteen patients presented with iHSRs to carboplatin (iHSR-Pl^pos^), and fourteen were tolerant (iHSR-Pl^neg^). The BAT median stimulation index (SI) values were 1.563 (range, 0.02–4.11; *n* = 11) and −0.28 (range −4.88–0.07, *n* = 11) in iHSR-Tax^pos^ and iHSR-Tax^neg^, respectively. The BAT median SI values were 4.45 (range, 0.1–26.7; *n* = 14) and 0 (range, −0.51–1.65; *n* = 12) in iHSR-Pl^pos^ and iHSR-Pl^neg^, respectively. SI levels were not associated with iHSR severity grading. Comparing BAT results in iHSR-Tax^pos^ and iHSR-Tax^neg^ showed the area under the receiver operator characteristic (ROC) curve to be 0.9752 (*p* = 0.0002). The cutoff calculated by the maximized likelihood ratio identified 90.91% of iHSR-Tax^pos^ patients and 90.91% of iHSR-Tax^neg^ patients. Comparing BAT results for iHSR-Pl^pos^ and iHSR-Pl^neg^ showed the area under the ROC curve to be 0.9286 (*p* = 0.0002). The cutoff calculated by the maximized likelihood ratio identified 78.57% of iHSR-Pl^pos^ patients and 91.67% of iHSR-Pl^neg^ patients. Most iHSR-Tax^pos^ patients for which ST was available (10/11) scored ST-negative and BAT-positive, whereas most iHSR-Pl^pos^ patients for which ST was available (14/14) scored both BAT- and ST-positive. This suggested the intervention of non-IgE-mediated mechanisms in iHSR-Tax^pos^ patients. Consistent with this view, an in silico molecular docking analysis predicted the high affinity of paclitaxel to the degranulation-competent MRGPRX2 receptor. This hypothesis warrants further in vitro investigations. In conclusion, the present study provides preliminary proof-of-concept evidence that this novel BAT has potential utility in understanding mechanisms underlying iHSRs to taxanes.

## 1. Introduction

Taxane- and platinum-based chemotherapy represents a cornerstone of treatment in gynecological [1,2] and breast [3,4] cancer. However, immediate hypersensitivity reactions (iHSRs) to these chemotherapeutic agents have become increasingly frequent because the sequential use of chemotherapy regimens and the incorporation of molecularly targeted treatments allows for the administration of more lines of treatment with taxanes and platinum salts and more cycles for each line [5,6,7,8,9,10,11,12]. The prevalence rates of taxane-induced iHSRs are around 6% and 4% in gynecological cancer [9] and breast cancer [13], respectively. iHSRs commonly occur in the first or second treatment cycle. The use of sufficiently prolonged premedication has partially decreased iHSR manifestations [9], whose incidence also depends on the type of taxane [10]. The prevalence of iHSRs to platinum salts is around 6–16% [5], and the risk increases suddenly with the sixth treatment cycle [11].

Various personalized desensitization protocols still allow cancer patients to continue taxane- and platinum-based chemotherapy after iHSR occurrence, avoiding the need to resort to other less effective or more toxic therapeutic approaches [12,14,15]. The choice of appropriate drug desensitization protocols in drug allergies is routinely based on iHSR severity grading, skin tests (STs), and laboratory tests, including serum-specific IgE assays and the basophil activation test (BAT) [16,17]. However, in the case of iHSRs to taxanes, no reliable laboratory in vitro tests—either serum-specific IgE assays or BAT—are available, and diagnosis and desensitization strategies rely on clinical evaluations and ST results [12].

The BAT is a well-established in vitro functional assay that uses flow cytometry to measure the modulation of basophil surface molecules associated with cell degranulation following stimulation with the anticipated allergen [18]. The BAT takes into account all characteristics of IgE and allergens and can contribute to the diagnosis of allergic diseases along with sensitization tests [18]. The BAT is widely used for the diagnosis of iHSRs to platinum salts [19,20,21], but the BAT is not part of diagnostic work-ups regarding iHSRs to taxanes. In the only study where the BAT was used to diagnose iHSRs to taxanes [22], diagnostic performance was poor owing to low sensitivity, likely because the in vitro detrimental effect of taxanes on basophil responsiveness went overlooked.

Thus, in the present study, we modified the BAT by introducing a novel acceptance criterion to address the effect of paclitaxel-induced detrimental activity on basophil responsiveness and to prevent the generation of false-negative results. This modified BAT was used to provide new insights into the mechanisms underlying iHSRs to paclitaxel in patients presenting with gynecological cancer and breast cancer. The present BAT concomitantly measures the two well-established CD63 and CD203c basophil degranulation markers, which are indicative of anaphylactic and piecemeal degranulation pathways, respectively [23,24]. We show that most patients with iHSRs to paclitaxel scored BAT-positive and had negative STs. To learn more about the underlying mechanism behind these findings, we performed in silico molecular docking analyses to predict the interaction of paclitaxel with the degranulation-competent MRGPRX2 receptor [25,26,27,28,29,30,31,32], a molecule expressed on mast cells and basophils [33,34] that can mediate non-IgE-dependent cell degranulation.

## 2. Materials and Methods

The Appendix A provides full details on iHSR classification, the ST procedure, blood sample collection, and chemotherapeutic agent preparation.

### 2.1. Study Population and Design

This was a not-for-profit, observational, single-center study approved by the internal institutional ethical committee (protocol BAT-KPRED-01/ID2581). Patients with gynecological cancer or breast cancer with a documented history of iHSRs to either paclitaxel (iHSR-Tax^pos^) or carboplatin (iHSR-Pl^pos^) were eligible and represented the test group. Patients with gynecological cancer or breast cancer who were treated and found tolerant to either paclitaxel (iHSR-Tax^neg^) or carboplatin (iHSR-Pl^neg^) represented the control group. Exclusion criteria included a recent (<3 weeks) iHSR and a recent (<2 weeks) exposure to steroids, anti-histamines, or immunomodulating agents. Informed consent was obtained from each patient prior to participation in the study.

### 2.2. Monoclonal Antibodies

Anti-CD45, anti-CCR3, and anti-CRTH2 monoclonal antibodies (mAbs) were used to positively identify the basophils in whole blood. Anti-CD3 mAb was used to exclude CCR3- and CRTH2-expressing T lymphocytes from analysis. The CD63 and CD203c markers were used to assess basophil degranulation because modifications to their expression during basophil activation can be complementary [23,24]. MAb specifications are reported in Appendix A. Optimal mAb concentrations were set in preliminary experiments, and a mAb cocktail was freshly prepared for each BAT.

### 2.3. BAT Procedure

The BAT was performed within 2 h of blood withdrawal. Blood was collected by venipuncture into an EDTA-containing vacutainer blood collection tube. Non-pyrogenic 3.5 ml polypropylene tubes were filled with 20 μL serial dilutions of drugs, 20 μL pf anti-FcεRI mAb or 20 μL of n-formyl-Met-Leu-Phe chemotactic peptide (fMLP) (positive controls from Flow CAST^®^, Bühlmann, Düsseldorf, Germany), and 20 μL of phosphate-buffered saline (PBS, OXOID, Hampshire, UK; negative control). Ca^2+^Activation Solution (50 µL, Beckman Coulter, Brea, CA, USA) was added because EDTA, which is a calcium-chelating agent, may hamper basophil responsiveness [18]. Whole blood (100 μL) was added to each tube. Samples were gently mixed and incubated at +37 °C for 20 min in a water bath. The mAb cocktail (44.6 µL/tube) was then added to each tube. Samples were gently mixed and incubated at RT in the dark for 20 min. “Fixation and lysis” solution (25 µL of fixation solution, IOTest 3-A07800, in 1 ml of Versalyse-A09777; Beckman Coulter) was added to each tube and immediately mixed. After 15 min at RT, samples were centrifuged at 500× *g* for 5 min, supernatants were discarded, and cell pellets were gently suspended via tapping. A fixation solution (300 µL) was added to each tube and immediately mixed. Samples were run in a CytoFLEX-LX^TM^ flow cytometer (Beckman Coulter). At least 1000 basophils were acquired for each sample run. The cytometer was flushed with PBS after each run to prevent carryover. A time parameter was acquired to check flow stability. The basophils were identified as CD45^+^CD3^-^CCR3^+^CRTH2^+^cells with low side scatter (SSC-A) signal, and changes in CD63 and CD203c marker expressions were used to measure basophil activation/degranulation. The flow cytometry gating strategy is depicted in Figure 1. An inhibitor of Bruton’s tyrosine kinase (BTK)-mediated signal transduction via FcεRI crosslinking, ibrutinib (IMBRUVICA^®^, Pharmacyclics LLC, Sunnyvale, CA, USA, and Janssen-Biotech, Inc. Titusville, NJ, USA), was used to investigate basophil degranulation mechanisms. Ibrutinib (100 nM) was added 2 min before incubation with drugs and controls.

### 2.4. Molecular Docking Analysis

The structure of the G-protein-coupled receptor MRGPRX2 in complex with C48/80 was retrieved from the Protein Data Bank (PDB code 7VV6 [35]) and imported into the Protein Preparation Wizard of Maestro (Schrödinger Release 2022-3: Maestro, Schrödinger, LLC, New York, NY, USA, 2020). The protein preparation process included correcting mislabeled elements, adding hydrogen atoms, assigning bond orders, and hydrogen bond optimization. The overall structure was then geometrically optimized to relieve steric clashes via energy minimization based on the OPLS4 force field [36]. A structural analysis of MRGPRX2 aimed at identifying druggable binding sites was performed using the SiteMap tool from Schrödinger (Schrödinger Release 2022-3: SiteMap, Schrödinger, LLC, New York, NY, USA, 2021). The program uses the OPLS4 force field to estimate the interaction energies of probes placed at all points along a three-dimensional grid that encompasses the entire protein [37,38]. The settings used involved the generation of at least 15 site points per reported site. The predicted SiteMap pocket was used to set up the grid (dimensions 20 × 20 × 20 Å) for virtual screening experiments. Five compounds (rocuronium, atracurium, docetaxel, paclitaxel, and C48/80) were docked into the generated grid using the Glide program in extra-precision (XP) mode [39]. Ligands were prepared and converted into 3D structures using Schrödinger’s LigPrep tool (Schrödinger Release 2022-3: LigPrep, Schrödinger, LLC, New York, NY, USA, 2021). Epik, implemented within, was used to assign likely protonation states at pH7 ± 2 and tautomers to each molecule. All docked compounds were rescored based on binding energy using the Prime/MM-GBSA method (Schrödinger Release 2022-3: Prime, Schrödinger, LLC, New York, NY, USA, 2021).

### 2.5. Evaluation of BAT Results and Statistical Analysis

BAT results were evaluated as stimulation index (SI) values for the following:(a)(CD63^+^ basophil percentage increase (SI-CD63%):%CD63^+^ basophils_stimulated_ − %CD63^+^ basophils_non-stimulated_.(b)CD63 mean fluorescence intensity (MFI) increase (SI-CD63MFI):[(CD63MFI_stimulated_ divided by CD63MFI_non-stimulated_) − 1].(c)CD203c^+^ basophil percentage increase (SI-CD203c%):%CD203c^+^basophils_stimulated_ − %CD203c^+^basophils_non-stimulated_.(d)CD203cMFI increase (SI-CD203cMFI):[(CD203cMFI_stimulated_ divided by CD203cMFI_non-stimulated_) − 1].

The highest SI values observed in any of the drug concentrations were used. Moreover, we verified whether combinations of the above SIs performed better than any single SI in deeming a BAT positive.

Positive predictive value (PPV) and negative predictive value (NPV) were used to assess the clinical relevance of the BAT in diagnosing hypersensitivity (iHSRs to culprit drugs as the gold standard) and tolerance (no iHSR after exposure to culprit drug as the gold standard):

PPV = true positives/(true positives + false positives)

NPV = true negatives/(true negatives + false negatives)

Receiver operating characteristic (ROC) curves were generated to assess the BAT’s diagnostic accuracy and to determine the best cutoff point, calculated as the value of the maximized likelihood ratio (GraphPad-Prism9.4.1). The non-parametric Mann–Whitney test was used to compare the distributions of two independent groups (GraphPad-Prism 9.4.1).

## 3. Results

### 3.1. General Characteristics of the Study Population

Between November 2019 and March 2023, 31 patients undergoing paclitaxel and carboplatin treatment were enrolled; 17 were iHSR-Tax^pos^ patients, and 14 were iHSR-Pl^pos^ patients. Most iHSRs to paclitaxel occurred at the first infusion, whereas most iHSRs to carboplatin occurred after the fifth infusion, as previously reported [40]. Among the 17 iHSR-Tax^pos^ patients, 14 patients had been treated with carboplatin and did not present iHSRs (iHSR-Pl^neg^). Among the 14 iHSR-Pl^pos^ patients, 11 patients had been treated with paclitaxel and did not present iHSRs (iHSR-Tax^neg^). No patient presented iHSRs to both drugs. A study flow diagram is reported in Appendix A. The clinical and allergological characteristics of the study patients are reported in Appendix A. All iHSR-Pl^pos^ patients (*n* = 14) had diagnoses of ovarian cancer. Among the 17 iHSR-Tax^pos^ patients, 4 patients had breast cancer, 10 patients had ovarian cancer, and 3 patients had endometrial cancer. The BAT was executed 3 to 5 weeks after iHSRs.

### 3.2. BAT for Diagnosis of iHSRs to Paclitaxel

Early experiments showed that high paclitaxel concentrations down-regulated CD63MFI and, to a lower extent, CD203cMFI in basophils (Appendix A). Because CD63 should be absent from resting basophils [23,24], the CD63 down-regulation was unexpected. We, therefore, postulated that basophils in the BAT were not in a fully resting condition, as commonly believed; rather, they were responding to non-specific activation inherent to the BAT culture conditions (i.e., 20 min of incubation at +37 °C). This phenomenon has usually been overlooked, as already highlighted [41,42]. To test this hypothesis, we cultured cells at a non-permissive temperature (+4 °C) in order to prevent any basophil spontaneous activation. To this end, CD63MFI and CD203cMFI were measured in basophils incubated at either +37 °C or +4 °C for 20 min in the absence of added stimuli. CD63MFI (Figure 2a) and CD203cMFI (Figure 2b) increased at +37 °C as compared with +4 °C.

This confirmed the induction of CD63 neo-expression caused by the culture conditions, explaining why paclitaxel was capable of inducing CD63MFI down-modulation. We concluded that high paclitaxel concentrations induce basophil distress, an unacceptable condition for an in vitro functional test such as the BAT. This prompted us to test a range of paclitaxel concentrations (0.015–0.12 μg/mL) in each BAT and to accept the results as indicative of either hypersensitivity or tolerance only at the drug concentrations that did not down-regulate CD63MFI. If no paclitaxel concentration fulfilled this acceptance criterion, the BAT was considered inconclusive, and the data were discarded.

The BAT met the acceptance criterion in 11 out of 17 iHSR-Tax^pos^ patients and 11 out of 11 iHSR-Tax^neg^ patients. The BAT results were evaluated using the diverse SI calculation methods reported in the Materials and Methods section. SI-CD63%, SI-CD203cMFI, and SI-CD63% + SI-CD203cMFI were significantly higher in iHSR-Tax^pos^ patients than in iHSR-Tax^neg^ patients (Figure 3a), while the other SI did not perform as well (not shown). The BAT’s performance in distinguishing iHSR-Tax^pos^ patients from iHSR-Tax^neg^ patients was then assessed using ROC curve analysis. The SI-CD63% values yielded an area under the ROC curve (AUC) of 0.9752 (*p* = 0.0002); the SI-CD203cMFI values yielded an AUC of 0.9008 (*p* = 0.0014); and the SI-CD63% + SI-CD203cMFI values yielded an AUC of 0.9752 (*p* = 0.0002) (Figure 3b).

According to the maximized likelihood ratio, the best cutoff values were as follows: SI-CD63% > 0.22, SI-CD203cMFI > 0.024, and SI-CD63% + SI-CD203cMFI > 0.1809 (Table 1). SI-CD63% > 0.22 and SI-CD63% + SI-CD203cMFI > 0.1809 both confirmed the diagnosis of iHSRs to paclitaxel in 10 out of 11 iHSR-Tax^pos^ patients and correctly indicated tolerance to paclitaxel in 10 out of the 11 iHSR-Tax^neg^ patients. SI-CD203cMFI > 0.024 was comparatively less informative (Table 1).

Interestingly, whenever basophil activation was triggered by paclitaxel, most basophils showed the enhanced expression of both CD63 and CD203c markers, and a minority of basophils showed the enhanced expression of either CD63 or CD203c. This indicated that the two degranulation pathways were simultaneously but also alternatively triggered by paclitaxel. Thus, although SI-CD63% and SI-CD63% + SI-CD203cMFI performed equally well in terms of diagnostic performance, we deemed paclitaxel-induced basophil degranulation to be better depicted by SI-CD63% + SI-CD203cMFI.

### 3.3. BAT for Diagnosis of iHSRs to Carboplatin

BAT has been demonstrated to be a reliable diagnostic assay for iHSRs to platinum salts [17,18,19] and was, therefore, used here to qualify the present BAT’s performance. Carboplatin did not affect basophil responsiveness at any of the tested concentrations (from 2.5 to 10.0 μg/mL; not shown). As with paclitaxel, the BAT results for iHSR-Pl^pos^ and iHSR-Pl^neg^ patients were evaluated as SI-CD63%, SI-CD203cMFI, and SI-CD63% + SI-CD203cMFI. The SI values obtained from the 14 iHSR-Pl^pos^ patients were higher than those of the 12 iHSR-Pl^neg^ patients using the three SI calculation methods (Figure 4a). The BAT’s performance in distinguishing iHSR-Pl^pos^ from iHSR-Pl^neg^ patients was then assessed using ROC curve analysis. Reliable results were obtained with all three SI calculation options (Figure 4b), with the SI-CD63% values yielding an AUC of 0.9286 (*p* = 0.0002), the SI-CD203cMFI values yielding an AUC of 0.9196 (*p* = 0.0003), and the SI-CD63% + SI-CD203cMFI values yielding an AUC of 0.9286 (*p* = 0.0002).

The best cutoff values evaluated using the maximized likelihood ratio were SI-CD63% > 0.665, SI-CD203cMFI > 0.115, and SI-CD63% + SI-CD203cMFI > 0.78 (Table 1). SI-CD63% > 0.665 and SI-CD63% + SI-CD203cMFI > 0.78 confirmed the diagnosis of iHSRs to carboplatin in 11 out of 14 iHSR-Pl^pos^ patients and correctly indicated tolerance to carboplatin in 11 out of 12 iHSR-Pl^neg^ patients (Table 1). These two SIs confirmed the diagnosis of iHSRs to carboplatin with 91.67% PPV and 78.57% NPV. As with the BAT results for the iHSR-Tax^pos^ patients, SI-CD63% + SI-CD203cMFI was selected to interpret BAT results.

### 3.4. Association between BAT Results and iHSR Severity Grading

We speculated that the extent of drug-induced basophil activation might correlate with iHSR severity grading. To this end, we compared the SI-CD63% + SI-CD203cMFI levels in patients with different iHSR severity grades and found that the two parameters were not associated (Appendix A). This might be due to the fact that pathophysiological mechanisms other than basophil degranulation contributed to the effector phase of iHSRs to paclitaxel and carboplatin. The time that elapsed from iHSR occurrence to BAT execution might have also contributed to the lack of association and warrants further investigations in a larger series of patients.

### 3.5. Relationship between BAT and ST Results

Among the 10 iHSR-Tax^pos^ patients with available STs, 8 patients had negative STs, in line with previous observations [12,40], and 9 patients had positive BATs (Appendix A). Conversely, among the 14 iHSR-Pl^pos^ patients with available STs, 13 patients had positive STs, in line with previous observations [8,40], and 11 patients had positive BATs (Appendix A). The high proportion of positive BATs in iHSR-Tax^pos^ patients presenting with negative STs suggested that IgE-mediated degranulation pathways are not involved in paclitaxel-induced basophil activation. To verify this hypothesis, we designed experiments in which the FcεRI-mediated basophil degranulation pathway was blocked with ibrutinib, a molecule reported to selectively inhibit IgE-mediated degranulation in human basophils [43]. Ibrutinib inhibited paclitaxel-induced basophil degranulation in iHSR-Tax^pos^ patients, irrespective of ST scoring, and in iHSR-Pl^pos^ patients as well. These results, however, cannot be taken as indicative of a selective blockade of IgE-mediated degranulation because ibrutinib unexpectedly inhibited non-IgE-mediated degranulation triggered by fMLP in some experiments (three out of eight) (Appendix A).

### 3.6. In Silico Molecular Docking of Taxanes to MRGPRX2 Receptor

The culprit drug binding to the pseudo-allergen receptor MRGPRX2 has been evoked to explain mast cell (MC) degranulation without the involvement of antibody priming in some drug-induced pseudo-allergic reactions [25,26,27,28,29,30,31,32]. The MRGPRX2 receptor has also been demonstrated to be expressed in human basophils [33,34]. Thus, we hypothesized that MRGPRX2 engagement with paclitaxel in MC and basophils could have provoked iHSR in the iHSR-Tax^pos^ patients presenting with positive BAT and negative ST results. Such a mechanism might also explain why most iHSR-Tax^pos^ patients experienced hypersensitivity during the first infusion, in line with early reports [10,40], since IgE-mediated iHSRs would have required previous exposure to the drug. We explored this possibility using structure-based molecular docking predictions of paclitaxel and docetaxel, another taxane molecule that has been described as provoking iHSRs in ST-negative patients [12,40]. The 48/80 compound (C48/80), one of the most potent MRGPRX2 agonists [25,26,35], and two neuromuscular blocking agents, rocuronium and atracurium—agonist ligands of the MRGPRX2 receptor that can induce iHSR [31,32]—were used as reference molecules. The determination of a druggable binding site for virtual screening was performed using the tool SiteMap in Maestro (Schrödinger). SiteMap ranks identified sites using two druggability assessment scores, SiteScore and Dscore, which characterize the binding site in terms of size, exposure to solvent, hydrophobicity and hydrophilicity, degree of hydrogen bond donation, and acceptance. Five druggable pockets were identified with comparable scores (Figure 5a), and Site2, corresponding to the C48/80 binding site in the MRGPRX2 cryo-EM structure (PDB code 7VV6), was selected for molecular docking calculations.

This pocket occupies approximately half the region of the extracellular surface, which comprises the N-terminus, the extracellular loop 2 (ECL2), ECL3, TM5, TM6, and TM7 and is lined by Phe170, Trp243, Trp248, Glu164, and Asp184 (Figure 5b), whose roles in signaling mechanisms have been described [35]. The set of C48/80, rocuronium, atracurium, paclitaxel, and docetaxel ligands (Figure 5c) was screened against the potential druggable pocket, Site2. First, the MRGPRX2 structure bound to C48/80 was used for docking validation (correctly identifying the binding site and scoring the receptor–ligand interactions). The ligand-docked pose was in close agreement with the experimentally determined position, as demonstrated by the root mean square deviation in the heavy atoms of the ligand (2.6 Å) and by the key contacts between C48/80 and the MRGPRX2 binding site. As shown in Figure 5d, in both the cryo-EM and modeled complexes, C48/80 forms hydrophobic contacts with Trp243 and Trp248 and charge–charge interactions with Asp184 and Glu164. The successful validation test prompted us to dock rocuronium, atracurium, docetaxel, and paclitaxel into the identified druggable pocket of the MRGPRX2 receptor. In the virtual screening workflow, post-docking Prime-MM-GBSA was used to estimate free binding energy and rescore the docking results. The predicted binding energy range for the five ligands was from −68.97 kcal/mol (atracurium) to −52.84 kcal/mol (rocuronium). The calculated binding energy of −58.10 kcal/mol for C48/80 suggests that all ligands may have interacted with the MRGPRX2 receptor. Free binding energies of −66.31 and −59.71 kcal/mol were predicted for paclitaxel and docetaxel, respectively. Of note, more favorable binding energy was shown by atracurium, paclitaxel, and docetaxel when compared with C48/80. An analysis of the predicted binding modes of the studied compounds is shown in Figure 5e.

## 4. Discussion

We optimized the BAT in the diagnosis of iHSRs to paclitaxel by introducing an acceptance criterion to exclude, in each BAT, paclitaxel concentrations that produce false-negative results by affecting basophil responsiveness. The present BAT distinguished iHSR-Tax^pos^ patients from iHSR-Tax^neg^ patients with an unprecedented 90.91% SE and 90.91% SP.

In the present study, most iHSR-Tax^pos^ patients scored BAT-positive. This finding is not in keeping with the only other study that has used BAT for the diagnosis of iHSRs to taxanes [22], in which only half of iHSR-Tax^pos^ patients scored BAT-positive. There are some possible explanations for this discrepancy. For example, the present study included iHSR-Tax^pos^ patients presenting with both negative and positive STs, and the control group comprised cancer patients exposed and tolerant to paclitaxel, whereas in that study, iHSR-Tax^pos^ patients with negative STs were excluded, and the control group comprised healthy subjects [22]. Most importantly, here, we introduced an acceptance criterion for BAT results consisting of the identification of paclitaxel concentrations not affecting basophil responsiveness. This prerequisite for accepting BAT results was underestimated in that study [22], in which, conversely, patients showing a taxane-induced reduction in CD63MFI were scored as BAT-negative.

That paclitaxel affected basophil responsiveness is not surprising. Paclitaxel has been reported to inhibit IgE-induced microtubule-dependent CD63-containing granule translocation in plasma membranes [44] by altering the polymerization dynamics of microtubules [45]. This implies that changes occurring in the expression of CD63 in paclitaxel-treated basophils in the BAT are the result of two opposing effects: on the one hand, the initiation of CD63-containing granule translocation in the cell membrane and, on the other hand, interference with CD63 translocation due to microtubule stabilization. This requires the testing of several paclitaxel concentrations in each BAT to identify which ones, if any, tilt the balance toward measurable CD63-containing granule translocation.

In the present study, most iHSR-Tax^pos^ patients presented with negative STs, consistent with the view that IgE-mediated allergies are not the only cause of iHSRs to paclitaxel [12,40,46]. We showed, for the first time, that most iHSR-Tax^pos^ patients scoring negatively in STs had positive BATs. This finding suggests that the basophils of iHSR-Tax^pos^ patients degranulate via non-IgE-mediated mechanisms. This is a novel observation because patients with negative STs were excluded from the only other study reporting BAT usage in iHSR-Tax^pos^ patients [22]. The involvement of non-immune degranulation pathways is consistent with the occurrence of iHSRs to taxanes during the first or second lifetime exposure in most patients [12,40]. The onset of iHSRs in the absence of documentable prior sensitization has been attributed to surfactants used in drug formulation [12,47]. However, one study reported basophil histamine release to be induced in vitro by paclitaxel alone but not by the surfactant Cremophor-EL [48], supporting the view that the taxane moiety may directly cause iHSR.

The MRGPRX2 receptor is a cell surface molecule shown to be involved in the IgE-independent mechanisms of basophil and MC degranulation in humans and to be engaged with various natural and xenobiotic compounds [30,31,33,49,50]. We hypothesized that the taxane molecule might have triggered basophil degranulation by binding to this receptor. Via molecular docking studies, we provided the first evidence of the taxane moiety binding to the MRGPRX2 receptor, with deduced binding energies of −66.31 and −59.71 kcal/mol for paclitaxel and docetaxel, respectively, similar to the binding energy values deduced for drugs such as atracurium and rocuronium–neuromuscular blocking agents, which have already been reported to provoke iHSRs [32]. According to these findings, the occupation of the MRGPRX2 receptor by taxanes may represent one of the non-IgE-mediated causative mechanisms of anaphylaxis in this class of chemotherapeutics. However, it is difficult to explain how it is possible that basophils in the BAT and not the MC in the ST were activated by paclitaxel through MRGPRX2 receptor occupancy. Yet, it is conceivable that high paclitaxel concentrations in the ST prevented MC degranulation by interfering with the polymerization dynamics of microtubules.

A major limitation of the present study is the small patient population size, which prevents the extendibility and generalizability of our findings. Future studies are warranted in order to definitively confirm the present observations in a larger validation set. Moreover, we do not have empirical evidence to support the prediction that taxanes and the MRGPRX2 receptor are suitable binding moieties. In vitro studies are needed to definitively prove MRGPRX2-dependent, taxane-induced basophil degranulation by using MRGPRX2 antagonists, inhibitors of MRGPRX2 downstream signaling [28], and MRGPRX2 selective silencing [25].

## 5. Conclusions

In conclusion, we described a modified BAT endowed with unprecedented SE and SP in order to complement ST results. This novel BAT may offer specialists insights into the mechanisms underlying the effector phase of iHSRs and possibly contribute to the personalized management of taxane iHSRs. We also hypothesized that the binding of taxanes to the MRGPRX2 receptor might contribute to the occurrence of iHSRs to taxanes. If confirmed, the management of patients with iHSRs to taxanes might take advantage of MRGPRX2 antagonists, as envisaged in preclinical studies [51,52,53].

## Figures and Tables

**Figure 1 cancers-15-05818-f001:**
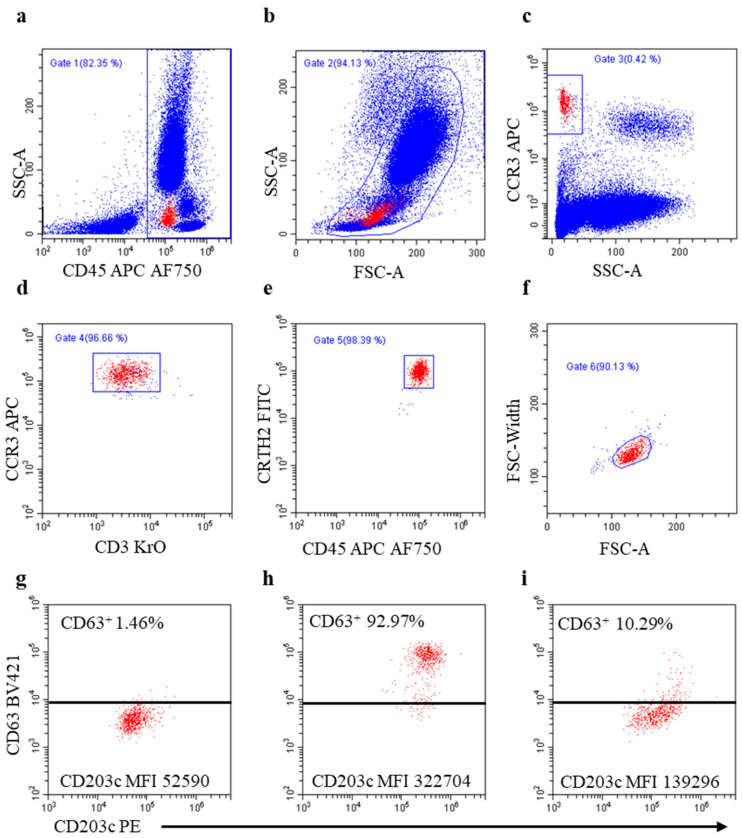
Gating strategy applied to analyze basophil activation in the BAT. (**a**) Selection of CD45^+^ leukocytes (Gate 1); (**b**) selection of leukocytes based on size (FSC-A) and complexity (SSC-A) (Gate 2); (**c**) identification of basophils based on CCR3 expression and low SSC-A signal (Gate 3); (**d**) exclusion of CCR3-expressing CD3^+^ lymphocytes (Gate 4); (**e**) selection of CRTH2^+^ basophils (Gate 5); (**f**) exclusion of doublets based on FSC-A signal pulse geometry gating (Gate 6); (**g**–**i**) basophil degranulation assessed as percentage of cells expressing CD63 and as median fluorescent intensity (MFI) of CD203c in the unstimulated control tube (**g**), in the anti-FcεRI mAb-stimulated control tube (**h**), and in a test tube containing carboplatin 10μg/mL (**i**). Events in red represent visualizations of the basophils in the final gate (**f**) (back-gating). Boundary position for evaluating percentage of CD63^+^ basophils in (**h**,**i**) was set on the unstimulated control (**g**).

**Figure 2 cancers-15-05818-f002:**
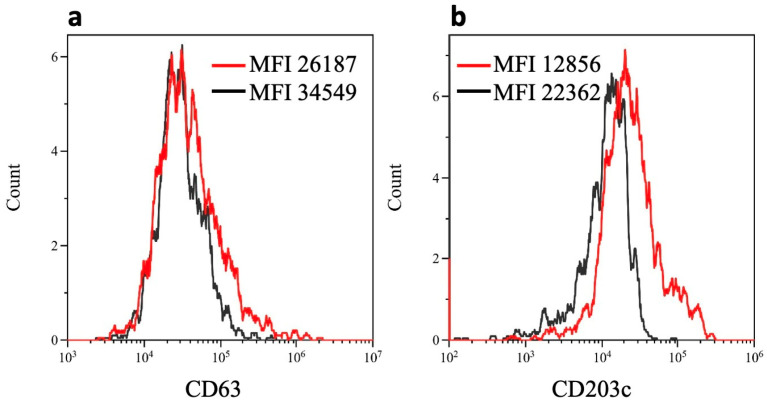
Effect of temperature on CD63 and CD203c mean fluorescence intensity (MFI) in basophils in the absence of stimulus. CD63MFI (**a**) and CD203cMFI (**b**) in basophils in blood samples stored at +4 °C (black lines) and +37 °C (red lines) for 20 min. One experiment representative of three is shown.

**Figure 3 cancers-15-05818-f003:**
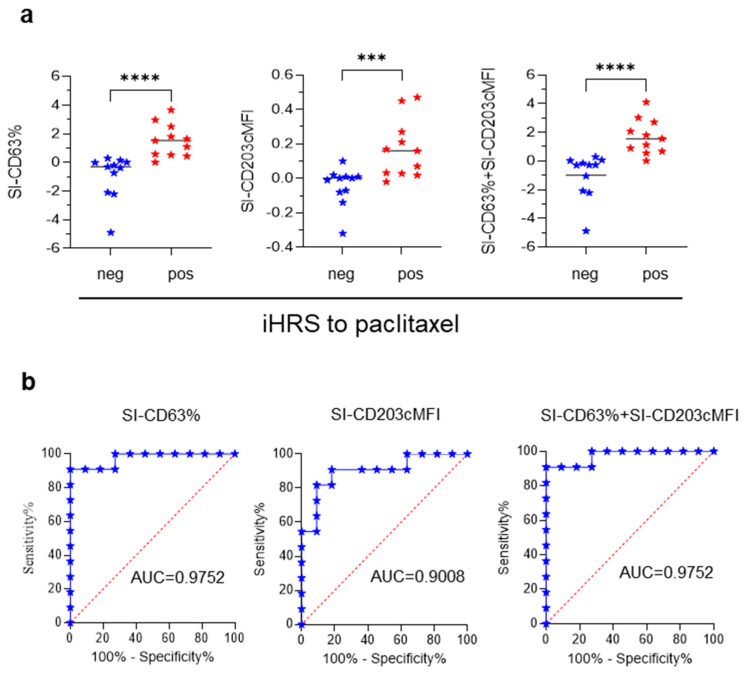
**The** BAT’s performance in recognizing patients with iHSRs to paclitaxel. (**a**) SI-CD63% ((**left**) panel), SI-CD203cMFI ((**middle**) panel), and SI-CD63% + SI-CD203cMFI ((**right**) panel) in the BAT on patients with negative (neg) and positive (pos) histories of iHSRs to paclitaxel. Horizontal lines represent median values. ***, *p* < 0.001 and ****, *p* < 0.0001 in Mann–Whitney non-parametric test. (**b**) Receiver operating characteristic (ROC) curve analyses. The area under the ROC curve (AUC) for the BAT in iHSR-Tax^pos^ and iHSR-Tax^neg^ patients is shown. (**Left**) panel: ROC curve of SI-CD63% values. (**Middle**) panel: ROC curve of SI-CD203cMFI values. (**Right**) panel: ROC curve of SI-CD63% + SI-CD203cMFI values.

**Figure 4 cancers-15-05818-f004:**
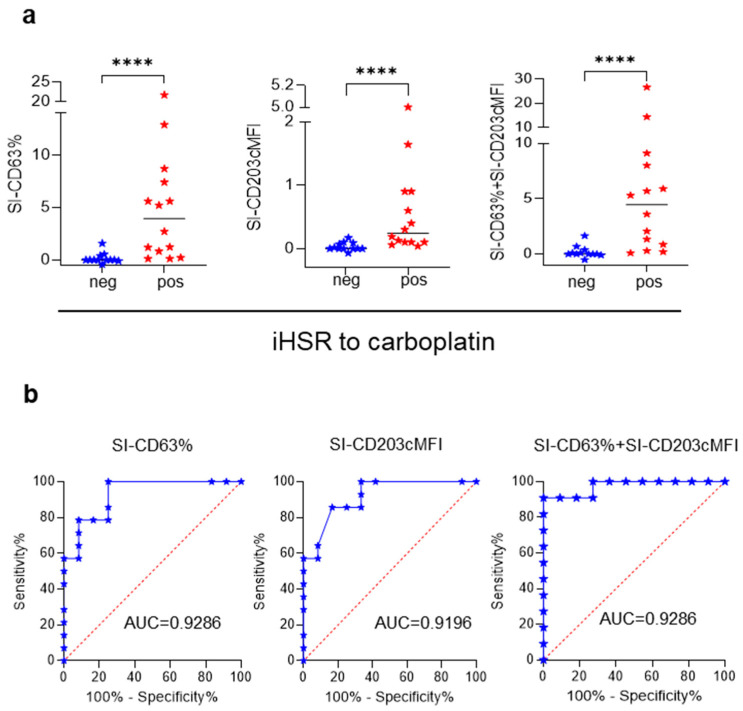
BAT performance in recognizing patients with iHSRs to carboplatin. (**a**) SI-CD63% ((**left**) panel), SI-CD203cMFI ((**middle**) panel), and SI-CD63% + SI-CD203cMFI ((**right**) panel) in the BAT on patients with negative (neg) and positive (pos) histories of iHSRs to carboplatin. Horizontal lines represent median values. ****, *p* < 0.0001 in Mann–Whitney non-parametric test. (**b**) Receiver operating characteristic (ROC) curve analyses. The area under the ROC curve (AUC) for the BAT in iHSR-Pl^pos^ and iHSR-Pl^neg^ patients is shown. (**Left**) panel: ROC curve of SI-CD63% values. (**Middle**) panel: ROC curve of SI-CD203cMFI values. (**Right**) panel: ROC curve of SI-CD63% + SI-CD203cMFI values.

**Figure 5 cancers-15-05818-f005:**
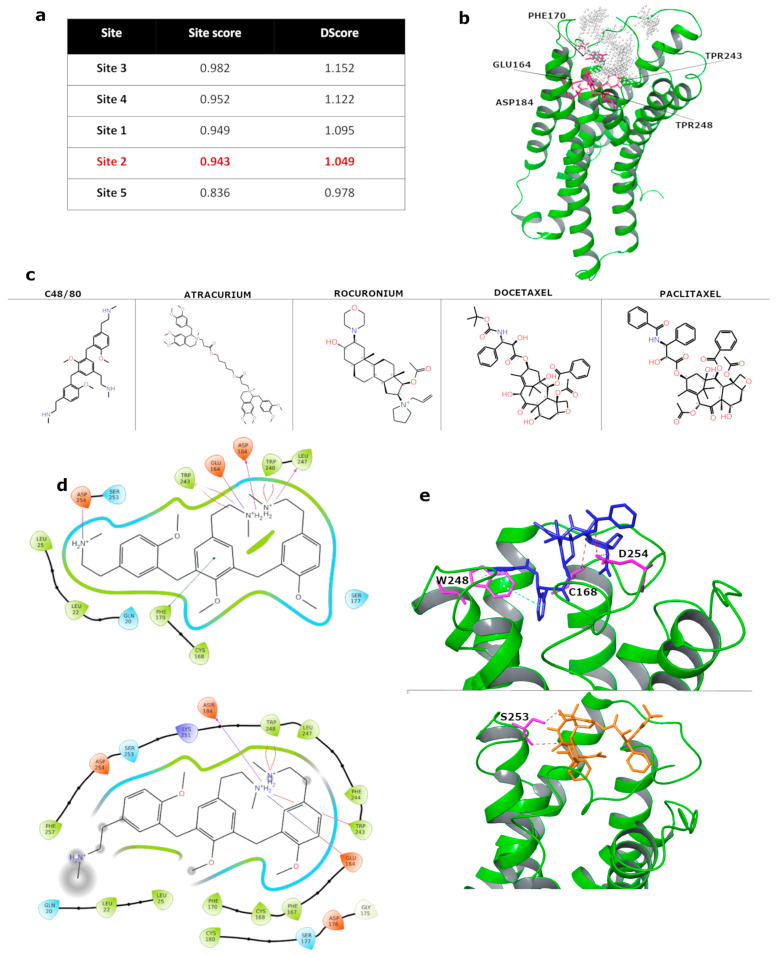
Molecular docking results. (**a**) Predicted SiteMap pockets on the MRGPRX2 surface. (**b**) Ligand binding sites in native MRGPRX2 as found by SiteMap. Protein is represented as green solid ribbons; the predicted SiteMap pocket is shown as white dots; and key residues are displayed as pink sticks. (**c**) Chemical structures of Compound 48/80, rocuronium, atracurium, paclitaxel, and docetaxel, tested in silico. (**d**) Two-dimensional ligand interaction diagram. Interactions of Compound 48/80 with the MRGPRX2 binding pocket in the cryo-EM ((**upper**) graph) and modeled complexes ((**lower**) graph). Purple arrows indicate hydrogen bonds; green and red lines indicate non-covalent interactions. Green represents hydrophobic residues, blue represents positively charged residues, red represents negatively charged residues, and cyan represents polar residues. (**e**) Predicted binding modes for paclitaxel ((**upper**) figure) and docetaxel ((**lower**) figure). The MRGPRX2 protein is shown as green ribbons. Key residues are displayed as pink sticks. Hydrophobic interactions and hydrogen bonds are shown as dashed lines in light blue and red, respectively.

**Table 1 cancers-15-05818-t001:** BAT performance in distinguishing patients with hypersensitivity to paclitaxel or carboplatin from tolerant patients.

Culprit Drug	SI Calculation Method	Cutoff Value	SE	SP	iHSR_pos_ Patients above Cutoff (Out of Total)	iHSR_neg_ Patients above Cutoff (Out of Total)
Paclitaxel	SI-CD63%	>0.22	90.91%	90.91%	10/11	1/11
Paclitaxel	SI-CD203cMFI	>0.024	81.82%	90.91%	9/11	1/11
Paclitaxel	SI-CD63% + SI-CD203cMFI	>0.1809	90.91%	90.91%	10/11	1/11
Carboplatin	SI-CD63%	>0.665	78.57%	91.67%	11/14	1/12
Carboplatin	SI-CD203cMFI	>0.115	64.29%	91.67%	9/14	1/12
Carboplatin	SI-CD63% + SI-CD203cMFI	>0.78	78.57%	91.67%	11/14	1/12

iHSR_pos_: patients presenting immediate hypersensitivity reactions to culprit drug; iHSR_neg_: tolerant patients; SI: stimulation index; MFI: mean fluorescence intensity; SE: sensitivity; SP: specificity.

## Data Availability

The datasets used and/or analyzed in the present study are available from the corresponding authors upon reasonable request.

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
