# Peer review of "A Modified Basophil Activation Test for the Clinical Management of Immediate Hypersensitivity Reactions to Paclitaxel: A Proof-of-Concept Study"

_cancers, 2023, doi:10.3390/cancers15245818_

Round 1

Reviewer 1 Report

Comments and Suggestions for Authors

A modified Basophil Activation Test for clinical management of immediate hypersensitivity reactions to paclitaxel

1-The authors are attempting to develop a test that can identify basophil activation of Taxanes using flow cytometry in order to detect iHSR as fast as possible.

The methods rely on CD63/CD203c upregulation and the authors assess the performance of the test using ROC and ML ratios. Also, paclitaxel concentrations were tested with this BAT method. The authors also mention the results obtained.

Overall, the abstract is clear.

Intro:

The authors start the manuscript by introducing taxanes and the possible iHSR that may arise in patients. The authors also provide stats about this and the premedication which may reduce the severity. Other aspects which may impact iHSR also get mentioned.

If this study is about gynaecological cancers, specifically, then the authors should give a specific section about the type of cancers they intend to cover in this study and what the iHSR stats look like for them. The intro is completely lacking this information while the methods mention breast and gyno cancer patients.

The next section talks about desensitisation and that specific and personalised methods may be used. Then they mention iHSR severity grading, ST and IgE tests. The authors could give one brief example of this, for example, what readings an example patient would get for these.

The main topic, BAT then gets mentioned which is flow-based methods for measuring basophil modulation but BAT is not sufficiently optimised for this and it may perform poorly for diagnosis. Since this is the core premise of the study, it would be useful, if the authors could give a brief example of this so that the readers can appreciate better how the current study has improved things.

In the last paragraph, the authors formulated their aims. The authors aim to explore the non-immune mechanisms involved in iHSR of taxane (since most BAT-positive test usually means ST-negative)

Protein docking was also investigated.

Overall, with the addition of the examples, the intro should be complete.

Methods:

The authors have used a flowchart to outline the patients included in this study (supplement). The patients with Tax or Pl and a history of iHSR were selected. 

The authors add breast and gyno patients, doesn’t this cause any heterogeneity? Since the iHSR rate in these cancer types was not mentioned and differentiated in the intro, it is not clear if this could be a source of heterogeneity in the results. Can the AUC and various ratios then be applied equally to them? Hopefully, the authors have split these groups later in the results section. The authors select iHSRneg patients for both drugs as control, which seems appropriate.

The next section mentions the type of antibodies used in this study to detect basophils and the basis of BAT tests and docking. The methods are generally detailed and well-explained.

The gating strategy is particularly useful since this helps future authors and investigators to select the same population or select other populations if need be.

Results:

In Figure 2, the author identifies that the basophils were not in a fully resting condition, but rather in a state of non-specific response which is why the MFI was different between 37 (biological) and 4 (low) degrees (in the presence of the stimulant from culture). One question would be whether it is possible that any other factors other than temperature could be a determinant here. So for example, is it possible to test the type of non-specific interaction it might be having?

The authors then tested concentrations of Pax and with a read-out of SI-CD63%, SI-CD203cMFI and SI-CD63%+SI-CD203cMFI they showed that these aspects are elevated in iHSR-Taxpositive patients and the performance assayed by ROC is acceptable, this is then followed by a table that shows the three readouts for 2 drugs and the cut-off imposed and the fraction of patients which lie above or below the cutoff.

This is generally good, but just bear in mind due to the difficulty in accumulating N numbers for this study, the authors need to be careful with their interpretation and state that larger numbers will enhance confidence in the results.

In many ways, the authors then use a “validation cohort” and test this for carboplatin and similar results are obtained. 

SI-CD63%+SI-CD203cMFI levels measured in the BAT of iHSR-Taxpos and iHSR-Plpos 293

patients did not associate with iHSR severity grading. My question is what is each SI individually (i.e., SI-CD63% and SI-CD203cMFI)? If not, the authors could speculate why. So, for instance, the readouts show the activation of basophils but not the extent to which they are activated. What other factors can play a role and are they testable?

The authors mention the time lapse since its execuation, do they mean the time before the test was conducted? Do the medical records of the patient give any keys as to what extent the severity was and why?

The result of the ibrutinib (S4) seems to be inconclusive, correct? In that IgE-mediated basophil activation was inhibited in Pax degranulation in iHSR+ patients regardless of ST status? Is there another way to test this? The compounds as you say even block the receptor (EceRI), so what if you tried to identify a compound that will competitively inhibit this aspect (binding and clocking the receptor), so the inhibition of IgE could be singled out to a large extent? Presumably, ibrutinib is using a different site on the receptor to block it, so maybe try and outcompete that with a reversible inhibitor.

I’m not sure about the point of looking at a receptor in mast cells when all the experiments so far were about basophils, yes you have linked them to paclitaxel, but it is still a big shift in the flow of the experiments.

Discussion:

What are the recommendations of these authors for the scientific community on the back of these experiments for BAT?

Comments on the Quality of English Language

Some minor editing required

Author Response

Thank you very much for taking the time to review this manuscript. Please find the detailed point-by-point responses below and the corresponding revisions/corrections highlighted/in track changes in the re-submitted files.

1. If this study is about gynaecological cancers, specifically, then the authors should give a specific section about the type of cancers they intend to cover in this study and what the iHSR stats look like for them. The intro is completely lacking this information while the methods mention breast and gyno cancer patients

Thank you for pointing this out. Information on statistics of iHRS in gynaecological and breast cancer patients has been added in the Introduction section. Information on cancer diagnosis in patients with iHSR to paclitaxel and to carboplatin in our study has been added in the Results section.

2. …Then they mention iHSR severity grading, ST and IgE tests. The authors could give one brief example of this, for example, what readings an example patient would get for these.As per the Reviewer’s request, below is the management of desensitization protocols for patients who have experienced iHRS to taxane, as performed at our institution. Desensitization procedure management is identical to what recommended in the papers of Picard M et al (2016) and Tsao LR et al (2021). Use of ST results and grading of iHSR to taxane both contribute to the choice of the desensitization protocols for reintroduction of taxanes.

3. …Since this is the core premise of the study, it would be useful, if the authors could give a brief example of this so that the readers can appreciate better how the current study has improved things.

We agree with this comment. We have better clarified in the Introduction section how present BAT might contribute to clarify the pathophysiology of the iHRS to taxanes.

4. The authors add breast and gyno patients, doesn’t this cause any heterogeneity? Since the iHSR rate in these cancer types was not mentioned and differentiated in the intro, it is not clear if this could be a source of heterogeneity in the results. Can the AUC and various ratios then be applied equally to them? Hopefully, the authors have split these groups later in the results section.

Thank you for this suggestion. It would have been interesting to explore this aspect. However, possible discrepancies in BAT diagnostic performance between gynecological and breast cancer patients could not be tested in this exploratory study due to the limited number of patients. This issue will be addressed in future studies with a larger population size.

5. In Figure 2, the author identifies that the basophils were not in a fully resting condition, but rather in a state of non-specific response which is why the MFI was different between 37 (biological) and 4 (low) degrees (in the presence of the stimulant from culture). One question would be whether it is possible that any other factors other than temperature could be a determinant here. So for example, is it possible to test the type of non-specific interaction it might be having?

We agree with this comment. It would have been interesting to explore this topic. In this study, culturing basophils at +4°C was just an expedient procedure for preventing any basophil activation, whatever the trigger was. We have clarified this concept in the Results section and added an appropriate reference.

6. … This is generally good, but just bear in mind due to the difficulty in accumulating N numbers for this study, the authors need to be careful with their interpretation and state that larger numbers will enhance confidence in the results.

We agree with this. This study limitation has been addressed throughout the manuscript and in the Conclusion section. The title of the manuscript has been modified accordingly.

7. SI-CD63%+SI-CD203cMFI levels measured in the BAT of iHSR-Taxpos and iHSR-Plpos 293 patients did not associate with iHSR severity grading. My question is what is each SI individually (i.e., SI-CD63% and SI-CD203cMFI)? If not, the authors could speculate why. So, for instance, the readouts show the activation of basophils but not the extent to which they are activated.

Thank you for pointing this out. We have tested for this and found that neither SI-CD63% nor SI-CD203cMFI associated with iHSR severity grading. Thus, we did not include this further analysis in the manuscript. As per the readouts of basophil activation, we have reported in the table below the extent range (min-max) of percentage of CD63+ basophils and of CD203c MFI observed upon exposure of basophils to the diverse stimuli in the whole study, for the Reviewer’s convenience.

Stimulus

CD63+ %

CD203c MFI

Unstimulated control

1.03-13.04

3,093-53,735

Carboplatin

1.34-23.14

4,818-199,748

Paclitaxel

1.1-6.11

18,901-62,750

FceR1

26.5-99.7

61,706-663,923

fMLP

15.1-59

97,671-245,050

8. What other factors can play a role and are they testable?

Thank you for raising this point. We cannot qualify our BAT as a quantitative in vitro assay, although we have frequently observed a dose-dependent increase in basophil activation in the BAT, mainly when using carboplatin (because using different paclitaxel concentrations caused two opposite effects, as pointed out in the manuscript; on the one hand, a toxic effect on basophils and on the other hand, basophil degranulation). Thus, lack of association between BAT results and iHRS severity grading may reflect the intrinsic characteristics of this semi-quantitative/qualitative test.

Perhaps most importantly, it can be speculated that basophil degranulation in the BAT does not recapitulate all possible pathophysiological mechanisms that evoked the iHSR. We have added a comment on this point in the Results section.

9. The authors mention the time lapse since its execution, do they mean the time before the test was conducted?

We meant the time lapse between iHRS occurrence and blood sample withdrawal for BAT execution. In our series of patients, BAT was performed after 3 to 5 weeks since iHSR. This has been now specified in the M&M section. A larger series of patients will make it possible to assess whether time to BAT execution would influence relation between BAT results and iHSR severity. We have rephrased the sentence in the Results section accordingly. 

10 Do the medical records of the patient give any keys as to what extent the severity was and why?

Thank you for having emphasized this point. No association has been found between comorbidities and iHSR severity grading in our series of patients. However, because of the small number of patients, we would rather not discuss this issue in the manuscript.

11. The result of the ibrutinib (S4) seems to be inconclusive, correct? In that IgE-mediated basophil activation was inhibited in Pax degranulation in iHSR+ patients regardless of ST status? Is there another way to test this? The compounds as you say even block the receptor (EceRI), so what if you tried to identify a compound that will competitively inhibit this aspect (binding and clocking the receptor), so the inhibition of IgE could be singled out to a large extent? Presumably, ibrutinib is using a different site on the receptor to block it, so maybe try and outcompete that with a reversible inhibitor.

You have raised an important point here. Ibrutinib results have been deemed inconclusive because ibrutinib, which was expected not to interfere with non-IgE-mediated basophil degranulation, did interfere with fMLP-induced basophil activation in 3 out of the 8 experiments performed. This unexpected observation prevented unequivocal interpretation of the ibrutinib-mediated blockade of paclitaxel-induced degranulation regardless of ST status. We have rephrased the paragraph for the sake of clarity. We agree on the suggestion of testing other compounds with selective inhibiting activity on FceR-mediated signal transduction. However, this important issue will be the focus of future research, due to the difficulty in enrolling patients in this kind of clinical setting.

12. I’m not sure about the point of looking at a receptor in mast cells when all the experiments so far were about basophils, yes you have linked them to paclitaxel, but it is still a big shift in the flow of the experiments.

We have proposed involvement of the MRGRPRX2 receptor in basophil degranulation by paclitaxel because this receptor, although originally described on mast cells, was later reported to be also expressed on basophils. We have added appropriate references on the MRGPRX2 receptor expression on basophils in the Results section.

13. What are the recommendations of these authors for the scientific community on the back of these experiments for BAT?

As per the Reviewer’s suggestion, we have highlighted the relevance of the BAT as a tool for contributing new insights to the understanding of mechanisms underlying iHRS to paclitaxel in the Conclusions section.

Reviewer 2 Report

Comments and Suggestions for Authors

The study described the development of skin hypersensitivity methods using a flow-cytometry-based Basophil Activation Test (BAT) for taxane (iHSR diagnosis) and for contributing to desensitization procedure choice. Patients with various reaction to paclitaxel were enrolled. Skin testing (ST) was performed (allergy diagnosis). Basophil activation was measured as CD63/CD203c up-modulation. One of the main findings is based on using In silico binding analysis of paclitaxel to MRGPRX2 receptor. The study is interesting and important, although there is a room for improvements,

11.  Abstract should be focused on the results, not just methods. For instance, what was the level of CD63/CD203c in patients with hypersensitivity to paclitaxel? Was it significantly different in the patinets with low sensitivity? What are the data for the tolerant patients to either paclitaxel (iHSR- Taxneg) or carboplatin (iHSR-Plneg) represented the control group? This data is missing in the Abstract. Considering that the population group is small, author should indicate that this is a pilot study.

22.  Abstract: it is unclear how many patients/samples were used in this study. That is a serious problem as it makes a confusing impression that this is a large study.

33. The other main problem, that the theoretical data which were received during in silico analysis were not than confirmed in vitro.

4. Introduction may indicate that authors also aimed to find the relevant receptor responsible for the hypersensitivity. They can indicate that mast cells can be involved etc. Introduction can be extended.

55. Limitations of this study are missing/should be indicated. At least authors should mention that all data require verifications in future in vitro/in vivo experiments.

66. In support of their data, authors may want to include a recent publication related to the current study by González-Díaz SN, Canel-Paredes A, Macías-Weinmann A, Vidal-Gutiérrez O, Villarreal-González RV. Atopy, allergen sensitization and development of hypersensitivity reactions to paclitaxel. J Oncol Pharm Pract. 2023 Jun;29(4):810-817. doi: 10.1177/10781552221080415.

77.     Discussion; author may extend the future perspectives and indicate that the hypersensitivity can be blocked using novel agents, see this study by Kumar M, Duraisamy K, Annapureddy RR, Chan CB, Chow BKC. Novel small molecule MRGPRX2 antagonists inhibit a murine model of allergic reaction. J Allergy Clin Immunol. 2023 Apr;151(4):1110-1122. doi: 10.1016/j.jaci.2022.12.805.

Comments on the Quality of English Language

good

Author Response

Thank you very much for taking the time to review this manuscript. Please find the detailed point-by-point responses below and the corresponding revisions/corrections highlighted/in track changes in the re-submitted files.

1. Abstract should be focused on the results, not just methods. For instance, what was the level of CD63/CD203c in patients with hypersensitivity to paclitaxel? Was it significantly different in the patinets with low sensitivity? What are the data for the tolerant patients to either paclitaxel (iHSR- Taxneg) or carboplatin (iHSR-Plneg) represented the control group? This data is missing in the Abstract. Considering that the population group is small, author should indicate that this is a pilot study.

We agree with these comments. Therefore, we have modified the Abstract accordingly. That our study is a pilot study has been clarified in the title.

2. Abstract: it is unclear how many patients/samples were used in this study. That is a serious problem as it makes a confusing impression that this is a large study

We agree. This information has been inserted in the Abstract.

3. The other main problem, that the theoretical data which were received during in silico analysis were not than confirmed in vitro.

We agree and we have acknowledged this in the Abstract and in the Discussion section.

4. Introduction may indicate that authors also aimed to find the relevant receptor responsible for the hypersensitivity. They can indicate that mast cells can be involved etc. Introduction can be extended.

We agree with this and have incorporated your suggestion in the Introduction section and in the Results section, as well.

5. Limitations of this study are missing/should be indicated. At least authors should mention that all data require verifications in future in vitro/in vivo experiments.

We agree and have described the limitations of our study in the Discussion section.

6. In support of their data, authors may want to include a recent publication related to the current study by González-Díaz SN, Canel-Paredes A, Macías-Weinmann A, Vidal-Gutiérrez O, Villarreal-González RV. Atopy, allergen sensitization and development of hypersensitivity reactions to paclitaxel. J Oncol Pharm Pract. 2023 Jun;29(4):810-817. doi: 10.1177/10781552221080415

We thank the Reviewer for the suggestion. We have quoted the publication by Gonzalez-Diaz et al. in the Discussion section.

7. Discussion; author may extend the future perspectives and indicate that the hypersensitivity can be blocked using novel agents, see this study by Kumar M, Duraisamy K, Annapureddy RR, Chan CB, Chow BKC. Novel small molecule MRGPRX2 antagonists inhibit a murine model of allergic reaction. J Allergy Clin Immunol. 2023 Apr;151(4):1110-1122. doi: 10.1016/j.jaci.2022.12.805.

We thank the Reviewer. We have incorporated your suggestion in the Conclusions and quoted the publication by Kumar et al., along with two other relevant papers.

Round 2

Reviewer 1 Report

Comments and Suggestions for Authors

The authors have provided a point-by-point response to all of my comments, thanks.